# Identification, Classification, and Expression Analysis of the *Triacylglycerol Lipase* (*TGL*) Gene Family Related to Abiotic Stresses in Tomato

**DOI:** 10.3390/ijms22031387

**Published:** 2021-01-30

**Authors:** Qi Wang, Xin Xu, Xiaoyu Cao, Tixu Hu, Dongnan Xia, Jianhua Zhu, Xiangqiang Zhan

**Affiliations:** 1State Key Laboratory of Crop Stress Biology for Arid Areas, College of Horticulture, Northwest A&F University, Yangling 712100, China; wangqi94@nwafu.edu.cn (Q.W.); xuxin961118@163.com (X.X.); cxy1025@nwafu.edu.cn (X.C.); htx0729@nwsuaf.edu.cn (T.H.); xiadongnan0110@163.com (D.X.); 2Department of Plant Science and Landscape Architecture, University of Maryland, College Park, MD 20742, USA

**Keywords:** tomato, triacylglycerol lipase (TGL), classification, phylogenetic analysis, gene expression patterns, abiotic stress responses

## Abstract

Triacylglycerol Lipases (TGLs) are the major enzymes involved in triacylglycerol catabolism. TGLs hydrolyze long-chain fatty acid triglycerides, which are involved in plant development and abiotic stress responses. Whereas most studies of TGLs have focused on seed oil metabolism and biofuel in plants, limited information is available regarding the genome-wide identification and characterization of the *TGL* gene family in tomato (*Solanum lycopersicum* L.). Based on the latest published tomato genome annotation ITAG4.0, 129 *SlTGL* genes were identified and classified into 5 categories according to their structural characteristics. Most *SlTGL* genes were distributed on 3 of 12 chromosomes. Segment duplication appeared to be the driving force underlying expansion of the *TGL* gene family in tomato. The promoter analysis revealed that the promoters of *SlTGL*s contained many stress responsiveness *cis*-elements, such as ARE, LTR, MBS, WRE3, and WUN-motifs. Expression of the majority of *SlTGL* genes was suppressed following exposure to chilling and heat, while it was induced under drought stress, such as *SlTGLa9, SlTGLa6, SlTGLa25, SlTGLa26*, and *SlTGLa13*. These results provide valuable insights into the roles of the *SlTGL* genes family and lay a foundation for further functional studies on the linkage between triacylglycerol catabolism and abiotic stress responses in tomato.

## 1. Introduction

The first generation of biodiesel was obtained through lipid research on plants and animals [1]. The high-energy-density of triacylglycerol or triacylglyceride (TAG) in plant oils, which is twice as much as carbohydrates, means that they represent a valuable source for biofuels. Plant oils can be transformed into biofuels by transesterification of the acyl and fatty acid (FA) [2]. TAGs, neutral lipids with three fatty acids esterified to a glycerol backbone, are an important compound with the highest density of carbon and energy in eukaryotes. Plants mainly biosynthesize plant oils (mostly as TAGs) in their seeds, which are the energy and carbon resource for seedling development [3]. Driven by the societal interests of obtaining renewable fuels and chemicals from plant oils, the utility and value of TAG are realized [4,5]. Several studies have indicated that gene manipulation can change plant oil storage in the vegetative organs [6,7]. Overexpression of the key limiting enzyme in Arabidopsis TAG synthesis, diacylglycerol acyltransferase 1 (AtDGAT1), has been shown to increase the TAG content in tobacco leaves [8]. Such findings highlight the great potential of research on TAG synthesis in plants for improving the efficiency of biofuel production.

In model plants, the metabolic network of TAG synthesis is well described, especially for Arabidopsis. For TAG catabolism, fatty acids can be cleaved off by triacylglycerol lipase (TGL) and further metabolized in peroxisomes through β-oxidation to yield acetyl-CoA [9]. Lipases are serine hydrolases defined as triacylglycerol acyl-hydrolases (E.C. 3.1.1.3) [9] and can hydrolyze triacylglycerol, diacylglycerol (DAG), and sometimes other substrates such as monoacylglycerol and glycerolipids [10]. All known lipases are characterized by a high turnover rate on a long chain TAG substrate [11,12,13]. All lipase structures share an α/β hydrolase fold [14]. The lipases’ lid domains are amphipathic, which have hydrophilic and hydrophobic residues, showing open or closed in organic and aqueous solvents [15,16,17].

There are reports about the functions of TGLs in biological processes in Arabidopsis [18,19,20,21]. AtLIP1 (LIPASE 1) affects seed germination by changing TAG composition at different stages, indicating that TAG content is important for seed germination [11,22]. AtSDP1 (SUGAR-DEPENDENT1 Lipase) is essential for early TAG breakdown in seedlings, because when *SDP1* is knocked out in Arabidopsis, the mutant line germinates abnormally [23]. Furthermore, when *AtSDP1* and its homolog *AtSDP1-LIKE* are both knocked out, the double mutant can degrade TAGs 5 days after seed germination [24], suggesting that other TGLs may still function during and post-seed germination in the double mutant. Together, these results suggest that TGLs are critical in seed germination and seedling development.

In recent years, a growing number of studies have shown that TAGs are involved in plant abiotic stress responses [25]. TAGs do not accumulate in plant vegetative tissues under normal growth conditions, whereas TAG synthesis is induced in response to abiotic stresses [26]. When Arabidopsis seedlings were exposed to high temperatures between 32–50 °C, TAGs accumulated in extra-chloroplastic compartments in both roots and shoots, indicating that TAGs may be needed for membrane remodeling during heat acclimation [27]. Such a response indicates that the expression of TAG production-related genes also changes under heat stress. Additionally, the expression levels of two synthetases in TAG biosynthesis, *AtLPAT4* and *AtLPAT5* (*LYSOPHOSPHATIDIC ACID ACYLTRANSFERASES 4* and *5*), have been shown to increase under nitrogen starvation conditions [28].

Tomato (*Solanum lycopersicum* L.) originated in tropical and subtropical areas, and it is now grown widely around the world [29,30]. Tomato is considered to be the leading vegetable crop, with a global yield of 182 million tons in 2018 (United Nations Food and Agriculture Organization statistics). China has become the main producer of processing tomato in the world. In natural environments, tomato is often exposed to various abiotic stresses during their life, which affects normal growth and productivity [31,32]. Thus, improving the abiotic stress tolerance of the tomato plant is important for the agricultural industry.

The high commercial value of tomato makes it an important model plant to study fruit development [29,31,33] and abiotic stress responses [34]. In previous studies, the lipase gene family has been analyzed in various species, however, no systematic investigation has been conducted for tomato. Given the significance of the function in seed germination, biofuel production, and abiotic stress responses, we focused on the *TGL* gene family in tomato to study their potential function in response to different abiotic stresses. We carried out phylogenetic tree analysis, gene structure prediction, characterization of chromosome distributions, analysis of gene duplication events and expression profiling under abiotic stresses of the *TGL* gene family, and subcellular localization of selected TGLs to provide new insights to uncover the potential biological function of the TGL family proteins. Thus, our work lays a strong foundation for further research on functions of triacylglycerol lipases in tomato.

## 2. Results

### 2.1. Genome-Wide Identification and Phylogenetic Tree Analysis of SlTGLs in Tomato

To obtain sequences of *TGL* genes in tomato, the amino acid sequences of AtLIP1 were used as queries to perform a NCBI BLAST [35] search on tomato genome sequences (ITAG4.0) [36]. A HMMER-BLASTP-InterProScan [37] strategy was used to search for genes encoding proteins containing the α/β hydrolase domain (PF04083.16), and we searched the tomato genome sequence database (https://solgenomics.net/organism/Solanu-m_lycopersicum/genome/) for genes that encode proteins belonging to SlTGLs. In total, we identified 129 *TGL* genes in tomato. A phylogenetic tree was constructed based on multiple sequence alignments of all potential TGLs from tomato and the two Arabidopsis TGL paralogs: AtLIP1 and AtMPL1 (MYZUS PERSICAE-INDUCED LIPASE 1) (Figure 1). All 129 SlTGLs were clustered into five subfamilies: Group a-e, including 26, 32, 22, 18 and 31 members, respectively. Names were assigned based on the chromosome ID in the subfamily from *SlTGLa1* to *SlTGLe31*. We then analyzed the characteristics of these SlTGL proteins including coding sequence (CDS), protein molecular weight (MW), isoelectric point (pI), length of protein sequences, trans-membrane domain, and subcellular localization (Appendix A). CDSs of *SlTGL* genes in all 5 groups varied from 369 bp (*SlTGLe20*) to 2,758 bp (*SlTGLd7*) in length, and the length of amino acid sequence ranged from 117 (SlTGLe8) to 829 (SlTGLd7) amino acids. The pI values of SlTGL proteins ranged from 4.73 (SlTGLd2) to 9.68 (SlTGLe24). Theoretical MWs of SlTGLs ranged from 13.06 kDa (SlTGLe8) to 90.76 kDa (SlTGLd7). The protein subcellular localization of SlTGL showed that 90 of 129 SlTGLs were predicted to be located in the chloroplast or cytoplasm, whereas the other was in the nucleus or endoplasmic reticulum. Furthermore, 35% and 55% of the SlTGLs in subfamilies a and c were trans-membrane proteins (Appendix A).

### 2.2. Analysis of Gene Structure and Conserved Motifs of SlTGLs

To study the gene structure of *SlTGL*s, their exon-intron organizations were analyzed using GSDS 2.0 [38]. The number of exons per gene ranged from 1 to 17 (Figure 2c), and the genes in the same subfamily showed highly similar exon numbers: 24 members (77%) in subfamily e contained 3 to 5 exons, 9 members (50%) in subfamily d had 5 to 6 exons, 73% of members in subfamily c contained 4 to 6 exons, 22 members (69%) in subfamily b had 3 to 7 exons, and in subfamily a, and 14 genes (50%) contained 7 to 9 exons.

To analyze conserved motifs in SlTGL proteins, ten putative motifs between 10 and 50 amino acids were predicted by using the MEME program (http://meme-suite.org/tools/meme) [39] and InterProScan [37]. Ten conserved motifs (Motif1-10, Appendix A) were detected among all SlTGL members (Figure 2). According to InterProScan annotation, Motif2 or Motif8 defines an α/β hydrolase domain (IPR029058), which accounts for fundamental function of triacylglycerol lipases, while the remaining 8 motifs were not yet functionally annotated by InterProScan. Same to the phylogenetic tree (Figure 1), all SlTGLs are categorized into the same five groups (Group a–e) based on the compositions of Motif 1-10 (Figure 2).

### 2.3. Chromosome Location and Synteny Analysis of SlTGLs

To determine the *SlTGL*s genomic distribution, we mapped them to the published tomato genome (Appendix A). Most *SlTGL*s were distributed on chromosomes 1, 2, and 3. There were 21 *SlTGL*s on chromosome 1, 25 *SlTGL*s on chromosome 2, 17 *SlTGL*s on chromosome 3, 6 on chromosome 4, 14 on chromosome 5, 7 on chromosome 6, 5 on chromosome 7, 9 on chromosome 8, 12 on chromosome 9, 8 on chromosome 11, and 3 *SlTGL*s each on chromosome 10 and 12. Gene families expand mainly through three possible methods: tandem duplication, segmental duplication, and whole-genome duplication [40]. To analyze the status of gene duplication, CDSs of all the *SlTGL* genes were analyzed with the BLASTp and MCScanX software [41]. The results showed 18 pairs of segmental duplication genes in the whole tomato genome, whereas no tandem duplication was identified (Figure 3). Therefore, compared with tandem duplication, segmental duplication appears to be the main driving force for the amplification of the *SlTGL* gene family. It is unclear whether these segmental duplication events are in the known segmental duplication regions of tomato genome [42]. It is worth noting that our segmental duplication analysis did not consider sequences flanking the *SlTGLs*. Therefore, our conclusion may have some limitations, especially when considering genomes of more than two relative species. To study the evolutionary selection of tomato *SlTGL*s family, we calculated the *SlTGL* gene pairs’ nonsynonymous (Ka), synonymous substitutions (Ks), and the Ka/Ks ratios (Appendix A). In this study, 83% of genes of duplicate pairs had a Ka/Ks ratio ranged from 0.1 to 0.3, and no pair of duplicated genes had Ka/Ks > 1 (Appendix A), suggesting that the *SlTGL* genes in tomato were under purifying selection.

### 2.4. cis-Elements in the Promoters of SlTGLs

*cis*-elements in promoter regions are involved in gene regulation. It is, therefore, important to study the *cis*-elements present in putative promoter regions of the *SlTGLs* identified in tomato. The 1.5-kb putative promoter regions of the 129 *SlTGL* genes were uploaded to the PlantCARE database [43]. In total, 57 *cis*-elements were identified (Appendix A). Apart from the conventional *cis*-elements that were detected in all promoters of *SlTGL*s, we divided the *cis*-acting elements into three groups: plant growth and development, stress responsiveness, and phytohormone responsiveness (Figure 4). In the plant growth and development group (291/1353), including endosperm expression (GCN4-motif and AAGAA-motif), shoot and root meristem expression (CAT-box), flowering (MRE, CCAAT-box, and AT-rich element), and zein metabolism (O2 site). The largest proportion was AAGAA-motif (33%). In the stress responsiveness group (342/1353), there were anaerobic induction (ARE, 21%), drought-inducibility (MBS, 11%), low temperature (LTR, 8%), stress (TC-rich repeats, 7%), stress (STRE, 3%), and wounding responsiveness (WRE3 and WUN-motif, 3% and 2%). A lot of *cis*-elements (720/1353) were clustered to the phytohormone responsiveness group, including ethylene (ERE), salicylic acid (TCA-element), Me-JA (MYC and TGACG-motif), auxin (TGA-element), and abscisic acid (ABRE). Especially, most components of *cis*-elements were MYC, ERE, and ABRE, which were relative to hormone-responsiveness, accounting for 29%, 22%, and 18%, respectively. These results suggest that complex regulatory networks might be involved in the transcriptional regulation of *SlTGL* genes.

### 2.5. Analysis of the Expression Patterns of SlTGLs in Different Organs

We used tomato RNA-seq data published in the Tomato Functional Genomics Database (TFGD) [44] to analyze the expression patterns of the *SlTGL* genes in different organs in all developmental stages of fruits. Transcripts for a total of 124 *SlTGL* genes were analyzed in the RNA-seq data, whereas the remaining 5 members (*SlTGLb2, SlTGLb7, SlTGLb17, SlTGLe25,* and *SlTGLe26*) were not annotated in TFGD. Based on the expression patterns, the hierarchical clustering of the remaining 124 *SlTGL*s is shown in Figure 5. In the newly clustered five groups (A–E), we found some genes expressed in specific organs, for example, 86% (19 of 26) of group A genes strongly expressed in roots, and the expression levels of more than half (7 of 13) of genes in group D in flowers were more than twice as high as those in buds. In contrast, group B had 27 genes, 78% (21 of 27) of them preferentially expressed in the buds. Most genes in group C tended to be expressed in breaker fruit. All genes in group E appeared to have reduced expression in the fruit development stage (fruit 1–3 cm, mature green, breaker, and 10 days after breaker). To better understand the expression profiles of the *SlTGLs*, we analyzed the expression patterns of five genes from group A (*SlTGLa6*, *SlTGLa9*, *SlTGLa13*, *SlTGLa25*, and *SlTGLa26*), which had high sequence similarity to *AtLIP1* and *AtMPL1* and had relatively different expression patterns in different organs, in different organs and fruit development stages by qRT-PCR analysis (Figure 7b). *SlTGLa6* had a high expression level in immature fruit, and three genes (*SlTGLa9*, *SlTGLa25*, and *SlTGLa26*) were preferentially expressed in leaves, whereas two genes (*SlTGLa9* and *SlTGLa26*) tended to express in breaker fruit. These results indicate that *SlTGL*s had different expression patterns in the fruit development stage, suggesting that they may be involved in fruit development in tomato.

### 2.6. Expression Patterns of SlTGLs under Abiotic Stresses

To better understand the role of *SlTGL*s in response to abiotic stresses, transcriptome analysis combined with the qRT-PCR assay was employed. Heat, cold, and drought were the main abiotic stress conditions in plants during the growth stage, so we determined the transcriptome of 4-week-old tomato plants in the cultivar of M82 under these three stress treatments by RNA-seq analysis. Based on the data analysis of our RNA-seq data results, we clustered the 129 *SlTGL*s into several groups based on their differential expression patterns (Figure 6). Under chilling treatment, 69% (9 of 13) genes in group I upregulated after 3 h treatment. Expression of 98% (86 of 88) genes in group II exhibited a decreased trend in response to 4 °C treatment, and almost all genes in group III exhibited an increasing trend in response to 4 °C cold treatment. Furthermore, we analyzed the expression patterns of five *SlTGL* genes (*SlTGLa6*, *SlTGLa9*, *SlTGLa13*, *SlTGLa25*, and *SlTGLa26*) under chilling stress (4 °C stress) by qRT-PCR analysis (Figure 7a). The results showed that these five genes had an increasing trend under chilling stress, indicating *SlTGL* genes may participate in chilling stress responses.

*SlTGL* genes exhibited two distinct patterns under heat stress, an upward and a downward trend (Figure 6b). Whereas 55% (6 of 11) of genes in group IV and 99% (70 of 71) of genes in group VI showed an obvious downward trend, 85% (23 of 27) of genes in group V had a clear upward trend under heat stress. Among them, we chose five genes (*SlTGLa6*, *SlTGLa9*, *SlTGLa13*, *SlTGLa25*, and *SlTGLa26*) to analyze their expression level after exposure to a 42 °C treatment for varying lengths of time. The transcript levels of *SlTGLa13* and *SlTGLa25* upregulated after heat treatment and peaked at 2 h; about 3-fold and 1.5-fold higher than 0 h, respectively. The transcript abundance of *SlTGLa9* was the highest, at approximately 17-fold, after 1 h heat treatment. There were no significant changes in the expression of other genes under heat stress. These results were consistent with the changes in RNA-seq data.

Under dehydration stress induced by PEG-8000 treatment, 91% of *SlTGL*s (except genes in group X) transcript levels were affected. All genes in group VIII displayed a significant downregulated trend, and all genes in group XI exhibited an upregulated trend (Figure 6). We analyzed the expression patterns of three downregulated genes (*SlTGLa6*, *SlTGLa13*, and *SlTGLa25*) and two upregulated genes (*SlTGLa9* and *SlTGLa26*) dehydration stress induced by PEG-8000 treatment by qRT-PCR analysis. The results showed that *SlTGLa6* and *SlTGLa25* transcription levels were suppressed by 0.2-fold and 0.5-fold, respectively, after PEG-8000 treatment for 24 h. *SlTGLa26* upregulated after PEG-8000 treatment and the expression level peaked at 6 h reaching about 4-fold. Therefore, *SlTGL*s may be of great importance in response to dehydration stress.

### 2.7. Subcellular Locations of SlTGLs

The subcellular localization of SlTGL proteins was predicted using WoLF PSORT [46] (Appendix A). The results showed that the SlTGL proteins were located in various cellular components. To validate these results, we selected *SlTGLa9* and *SlTGLa13* which transcripts specifically expressed in some organs or were highly induced under abiotic stresses for a transient expression assay by using an expression system in tobacco (*Nicotiana benthamiana*) leaves (Figure 8). In WoLF PSORT predictions, SlTGLa9 is mainly located in the nucleus and SlTGLa13 may be located in the mitochondria, cytosol, and tonoplast. When transiently expressed in tobacco, the SlTGLa9-GFP fusion protein signal was merged with the mCherry-RFP fusion protein signal (served as a positive control for nucleus localization), indicating that SlTGLa9-GFP is located in the nucleus. The SlTGLa13-GFP fusion protein signal was not merged with the mCherry-RFP fusion protein signal, indicating that SlTGLa13 is not located in the nucleus. The SlTGLa13-GFP fusion protein appears to be detected in various subcellular structures. These results were consistent with the WoLF PSORT predictions for subcellular localizations of SlTGLa9 and SlTGLa13.

## 3. Discussion

TAGs play an important role in seed germination and are a major source for biofuel production. As the dominant enzymes for triacylglycerol catabolism, the underlying mechanisms of TGLs function need to be further investigated. A previous study in Arabidopsis showed that AtLIP1 participates in seed germination and has a lipase ability when expressed in the baculovirus system [11]. Tomato as a globally important food crop that is widely cultivated, there is a great economic demand to increase the stress resistance and fruit quality of tomato plants. In this study, we aimed to present a landscape of tomato TGLs, by conducting analyses of phylogenetic trees, conserved motifs, gene structure, and putative *cis*-elements in promoter regions, chromosome location, and gene expression profiles under different abiotic stresses.

In this study, we carried out a genome-wide analysis for *SlTGL*s in tomato and identified 129 total *SlTGL*s. Based on their protein sequences, we clustered all proteins into five groups, which had high similarity within each subfamily. Compared with *TGL*s from Arabidopsis, the tomato had more family members. Expansion of the gene family was observed, which commonly occurs by gene duplication over a long period and requires the force of environmental and biological factors [40,47,48]. Gene duplication and syntenic analysis suggest that segmental duplication is the major force for diversity in the *SlTGL*s family (Appendix A). Segmental duplication allows the preservation of the core functional group but creates divergence in the form of duplicated genes. The Ka/Ks ratio reflects that the *SlTGL* gene family evolved under the influence of purification (Appendix A). The novel motif analysis, based on the amino acid sequences, provides a pattern of related sequences in accordance with the position-dependent letter probability matrices [39,49]. The distribution frequently identified motifs, indicating the structural and functional similarity among *TGL* proteins in tomato. Motif2 is the most conserved structure and is the homology with the α/β hydrolase, which is consistent with the common α/β hydrolase fold of bacteria and animals [9]. Each subfamily was found to have different domains, possibly accounting for their different roles. For example, 82% of genes (18 of 22) in subfamily c had two α/β hydrolase domains, perhaps indicating their strong function in the hydrolysis of TAGs. TAGs are stored as neutral lipids within lipid droplets (LD) inside plastids. The degradation of LD is initiated with the degradation of LD-associated proteins, in which subsequently, peroxisome can interact with the LD and deliver AtSDP1 into the LD surface. TGLs, in association with other lipases, hydrolyzes the TAG into FAs and glycerol [25]. By predicting the transmem brane domain, we found that 35% and 55% of members in subfamily a and c, respectively. The TGLs play a prominent role in this reaction by catalyzing the initial step of cleaving TAG to DAG and FA. TAG is usually stored in lipid dr oplets or plastids which have monolayer or bilayer membrane structures, thus a transmembrane domain is crucial for TGLs to perform hydrolysis. A considerable number of genes in subfamily a and c have transmembrane domains, indicating that these two subfamilies play an important role in the hydrolysis of TAG.

The results of promoter analysis showed that various abiotic related *cis*-elements exist in the promoter regions, indicating *SlTGL*s may participate in various abiotic stress responses (Figure 4). We utilized the published transcriptome database (TFGD) to obtain a comprehensive understanding of *SlTGL*s expression patterns within the different plant organs. The different groups showed distinct expression patterns, for example, almost all genes in group A upregulated in the roots and the expression level of group C and E genes changed notably in parallel with fruit development, suggesting these two groups may be crucial for fruit development. Indeed, lipid composition is an important feature for fruit in the development stage [50,51], with lipophilic compounds in tomato fruits participating mainly in signaling, membrane structure, and development [52,53]. The diverse expression levels of *SlTGL* family genes in different organs or fruits at different stages, which was also confirmed by the in-house qRT-PCR analysis (Figure 7), indicates that these genes may have specific functions in the various developmental stages of tomato organs.

Various abiotic stresses including drought, heat, and chilling, can induce numerous stress response mechanisms, and activate related genes required for stress resistance. From the promoter analysis result, all *SlTGL*s promoters contained at least one stress responsive *cis*-element (HSE, TC-rich, MBS, and ARE), implying they might have a potential role in response to abiotic stress. To reveal the response of *SlTGL*s in abiotic stress treatments, transcriptomes of gene expression under various abiotic stresses were analyzed. The expression level of group II genes was suppressed under 4 °C treatment, while the genes in group I upregulated under 4 °C treatment (Figure 6a). Under heat stress, genes in group IV were induced, while the members in group VI were suppressed (Figure 6b). In the expression profiles of drought stress, genes in group VIII downregulated and group IX genes showed a pattern of upregulation (Figure 6c). To further test our hypothesis, we selected five genes for qRT-PCR verification. It can be seen from Figure 7a that the expression of these five genes is induced by abiotic stresses, indicating that they may be involved in the response to abiotic stresses. The expression patterns in different abiotic stress strengthened the evidence that *SlTGLs* participate in the response of tomato plants to abiotic stress conditions. Consequently, we speculated that these genes might have an important function in abiotic stress resistance, based on the lipid composition changes under heat stress in Arabidopsis [27].

In conclusion, some of the *SlTGLs* gene family are responsive to abiotic stresses and they may participate in plant abiotic stress responses although in planta functional studies will be required to validate their roles in such biological processes. Our present work enhances the understanding of the involvement of *SlTGL*s in response to changes in the natural environments, and provides the candidate genes for further functional studies.

## 4. Materials and Methods

### 4.1. Identification of SlTGLs in Tomato

To identify all the *SlTGL*s in the tomato genome, the LIP and MPL1 proteins reported in Arabidopsis were used as queries in the BLASTp program [35] against the latest *S. lycopersicum* whole proteome file in ITAG Release 4.0 from the Sol Genomics Network website (SGN, https://solgenomics.net/) [36]. HMM profiles of related SlTGL domain sequences (PF04083.16) were downloaded from the Pfam database (http://pfam.xfam.org/) [54] to identify the SlTGL domain genes using the HMMER software (version 3.0). All the redundant sequences were removed, and the remaining sequences were analyzed to confirm the presence of SlTGL domain by submitting them to the SMART database (http://smart.embl-heidelberg.de/) [55] and Pfam database [54]. Each sequence was then inspected manually. Collinearity and gene duplication events were analyzed using the BLASTp and MCScanX software package [41] with *SlTGL* coding sequences (CDSs). Protocol for collinearity analysis was similar to that reported by Song et al. [42] with slight changes.

### 4.2. Phylogenetic Tree Construction and Structural Analysis of SlTGLs

To study the phylogenetic relationship of the tomato *SlTGL* genes, a multi-sequence alignment was constructed with the MEGA X software [56]. A phylogenetic tree based on the alignment was constructed with the NJ method with 1000 bootstrap replicates. The exon/intron structures of the tomato TGL genes were determined with the online program Gene Structure Display Server (http://gsds.cbi.-pku.edu.cn/) [38].

### 4.3. cis-Element Prediction for SlTGLs Promoters

The sequences of *SlTGL* genes were used as queries in BLASTN searches against the tomato genome data (SL4.0) at the SGN website. The promoter sequences (1.5 kb upstream of 5′-UTR) of all the annotated *SlTGL* genes were submitted to the PlantCARE database [43] for cis-element prediction.

### 4.4. Expression Analysis of SlTGLs in Different Tissues and Fruit Developmental Stages of Tomato

The expression data for *SlTGL* genes in four tissues (bud, flower, leaf, and root) and six fruit developmental stages (1 cm fruit, 2 cm fruit, 3 cm fruit, mature green, breaker, and breaker after 10 days) were retrieved from the Tomato Functional Genomics Database (TFGD, http://ted.bti.cornell.edu/) [44]. The expression profiles, as fragments per kilobase per million reads (FPKM), of the tomato *SlTGL* genes were extracted with Linux, clustered, and drawn with the heatmap package in the TBtools software [45] with Euclidean distances and the complete linkage method of hierarchical clustering.

### 4.5. Analysis of SlTGL Genes in RNA-Seq Data

The tomato (*S. lycopersicum* L. cv. M82) plants were cultured in a greenhouse. Four-week-old tomato plants were treated with different abiotic stress. For chilling stress, plants were incubated under 4 °C, samples were collected at 0, 3, 12, 24 h. Plants were incubated under 42 °C for heat stress, samples were collected at 0, 2, 4, 12, 24 h. For drought treatments, a water-withholding experiment was performed for 5 days and samples were collected at 0, 1, 2, 3, 4, 5 days post the stress treatments. Leaves were collected as samples, frozen in liquid nitrogen, and stored at −80 °C. Residual genomic DNA was removed from the total RNA with the Turbo DNA-Free kit (Ambion, Shanghai, China) following the manufacturer’s instructions. The RNA extraction and next-generation sequencing with Illumina HiSeq2500 were performed in the Novogene (Beijing, China) and Biomarker (Beijing, China) Technologies Corporation. RNA-Seq data were analyzed by the TopHat and Bowtie2 programs. We subsequently used the Cuffdiff, a program within Cufflinks (http://cufflinks.cbcb.umd.edu/), for differential gene expression analysis. We have submitted the RNA-seq data to NCBI Gene Expression Omnibus (GEO) repository (accession numbers: SAMN14996375-14996413).

### 4.6. Plant Materials and Treatments for qRT-PCR Analysis

The tomato (*S. lycopersicum* L. cv. Alisa Craig) plants were cultured in a greenhouse or growth chambers. Three-month-old tomato plants were used to analyze the transcript levels of the *SlTGL* genes in different tissues. The roots, stems, leaves, flowers, and different fruit growth stages were collected for RNA extraction. To analyze the transcript levels of the *SlTGL* genes after different abiotic stress treatments, 2-week-old tomato seedlings were cultured in ½ Murashige and Skoog (½ MS) solid medium containing 200 mmol/L NaCl, and 20% polyethylene glycol (PEG, average molecular weight 8000). The seedlings were incubated at 4 °C and 40 °C to induce cold or heat stress, respectively. In all three treatment groups, the whole plants were collected after treatment for 0, 0.5, 1, 2, 4, 6, 12, or 24 h. All the collected samples were frozen in liquid nitrogen and stored at −80 °C before RNA extraction, cDNA synthesis, and quantitative expression analysis. Total RNA was extracted with the TRNzol Universal Total RNA Isolation Kit (Tiangen, Beijing, China), according to the manufacturer’s protocol. First-strand cDNA was synthesized from 1 µg of total RNA with the HiScript II 1st Strand cDNA Synthesis Kit (Vazyme, Nanjing, China). Real-time PCR was performed as described previously [57]. All qRT–PCR experiments included two technical replicates and three independent biological repetitions. *SlACTIN7* gene was used as a reference gene. The relative gene expression values were calculated using the 2^−∆∆Ct^ method. Gene expression values were log_2_ transformed, and heatmaps were generated using the heatmap package in the TBtools software [45]. The gene-specific primers (Appendix A) were designed, according to the CDSs of genes.

### 4.7. Sub-Cellular Localization

To determine the subcellular localization of the SlTGLs, the full-length coding sequences of two *SlTGL* genes were amplified from a tomato cDNA library constructed with 2-week-old young seedlings. Primers used for the PCR amplifications are listed in Appendix A. The amplified sequence was fused with GFP in an expression vector (pBI121-GFP) driven by the cauliflower mosaic virus (CaMV) 35S promoter. After the agrobacterium (strain GV3101) were grown to an OD_600_ of 0.6 at 28 °C, it was resuspended in infiltration medium and infiltrated into tobacco leaves, cultured in the dark for one day, and cultured under light for one additional day. Nuclei were visualized by co-transformation of a red fluorescent protein (RFP) fused with the nucleus marker mCherry. The fluorescence signals of SlTGL-GFP and mCherry protein in tobacco leaves were detected using the Olympus BX53 microscopy system (Olympus Microsystems, Tokyo, Japan) 48 h after the infiltration.

## Figures and Tables

**Figure 1 ijms-22-01387-f001:**
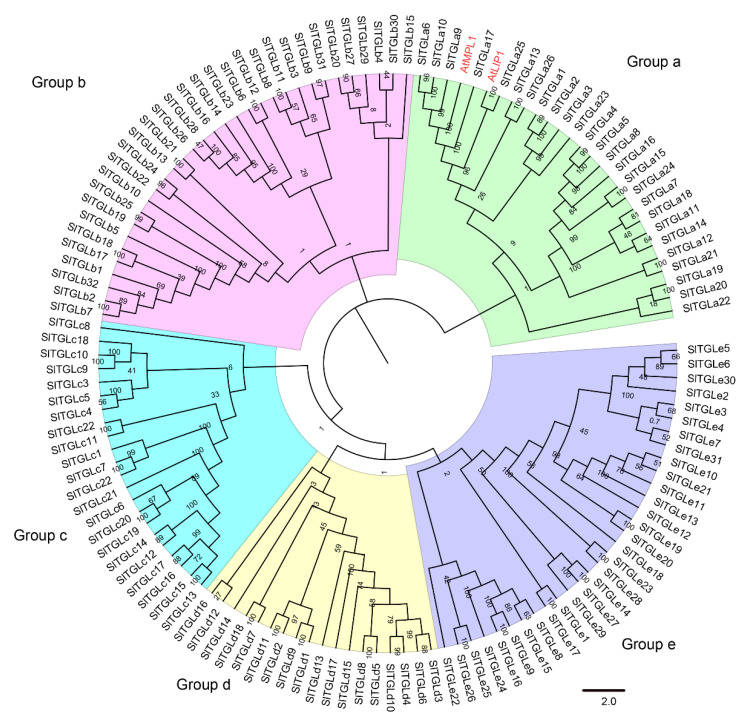
Phylogenetic relationships of SlTGLs from tomato (*S. lycopersicum* L.) and AtLIP1 and AtMPL1 from Arabidopsis. Phylogenetic analysis was performed using the neighbor-joining method with 1000 replicates. The TGL proteins were clustered into five groups. The branches of different subfamilies (group a–e) are marked using different colors. Homologous sequences in Arabidopsis are highlighted in red.

**Figure 2 ijms-22-01387-f002:**
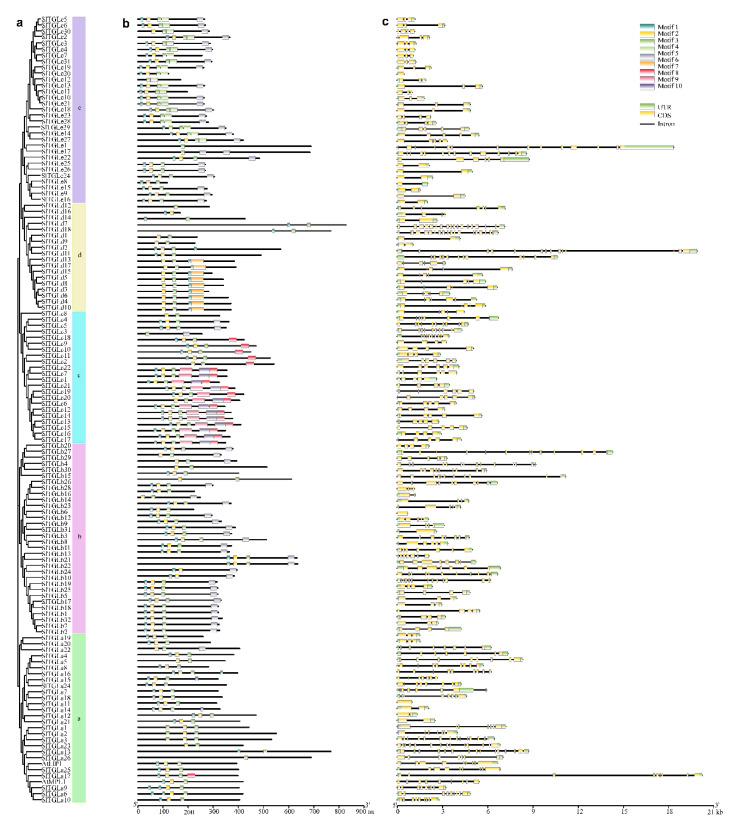
Phylogenetic relationships, architecture of conserved protein motifs, and gene structure of *SlTGLs* in tomato. (**a**) A phylogenetic tree constructed based on domain amino acid sequence of SlTGL proteins using MEGA X software. Groups a-e in different clusters were highlighted in different colors consistent with Figure 1. (**b**) Motif composition of SlTGL proteins. The motifs (Motif1~10) are displayed by using different colors. (**c**) The exon-intron structure of *SlTGL* genes. Green boxes indicate untranslated 5′ and 3′ regions of *SlTGL*s; yellow boxes represent exons; Introns were marked by black lines.

**Figure 3 ijms-22-01387-f003:**
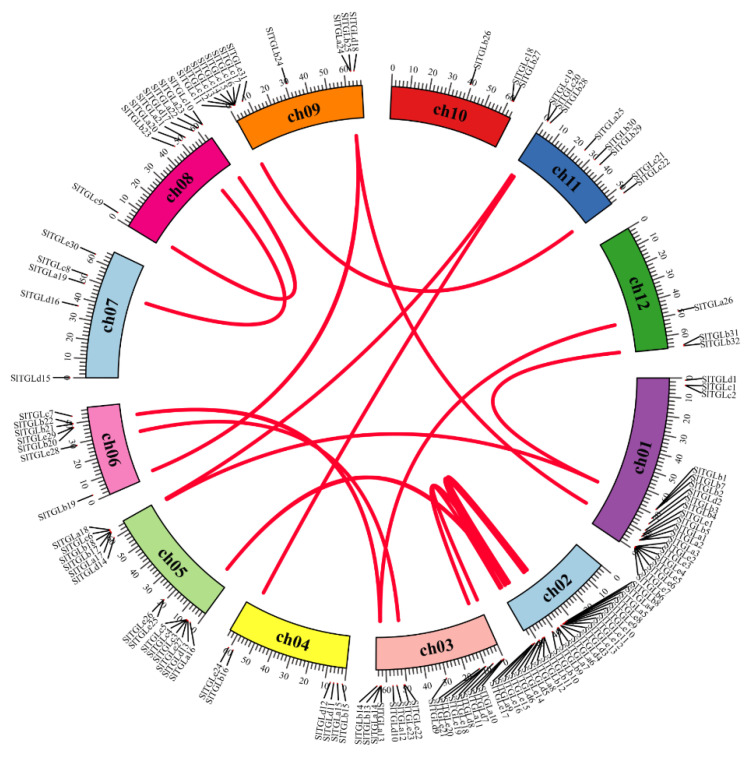
Distribution and segmental duplication of *SlTGL* genes in tomato. The panel shows the 12 chromosomes using a circle; red lines connecting homologous genes; chromosome numbers are marked inside of the circle.

**Figure 4 ijms-22-01387-f004:**
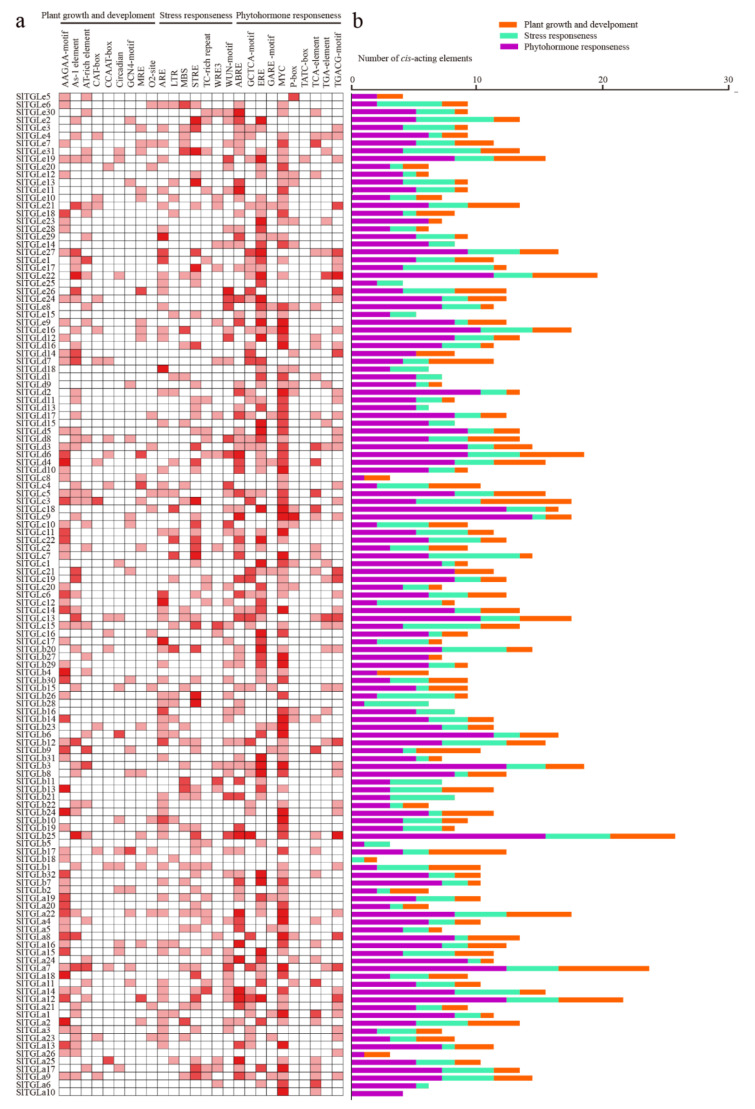
Information of *cis*-acting elements in putative promoter regions of *SlTGL*s. (**a**) The gradient colors in the red grid indicate the number of *cis*-acting elements in putative promoter regions of *SlTGL*s. (**b**) The different colored histogram indicates the *cis*-elements comportment in each category.

**Figure 5 ijms-22-01387-f005:**
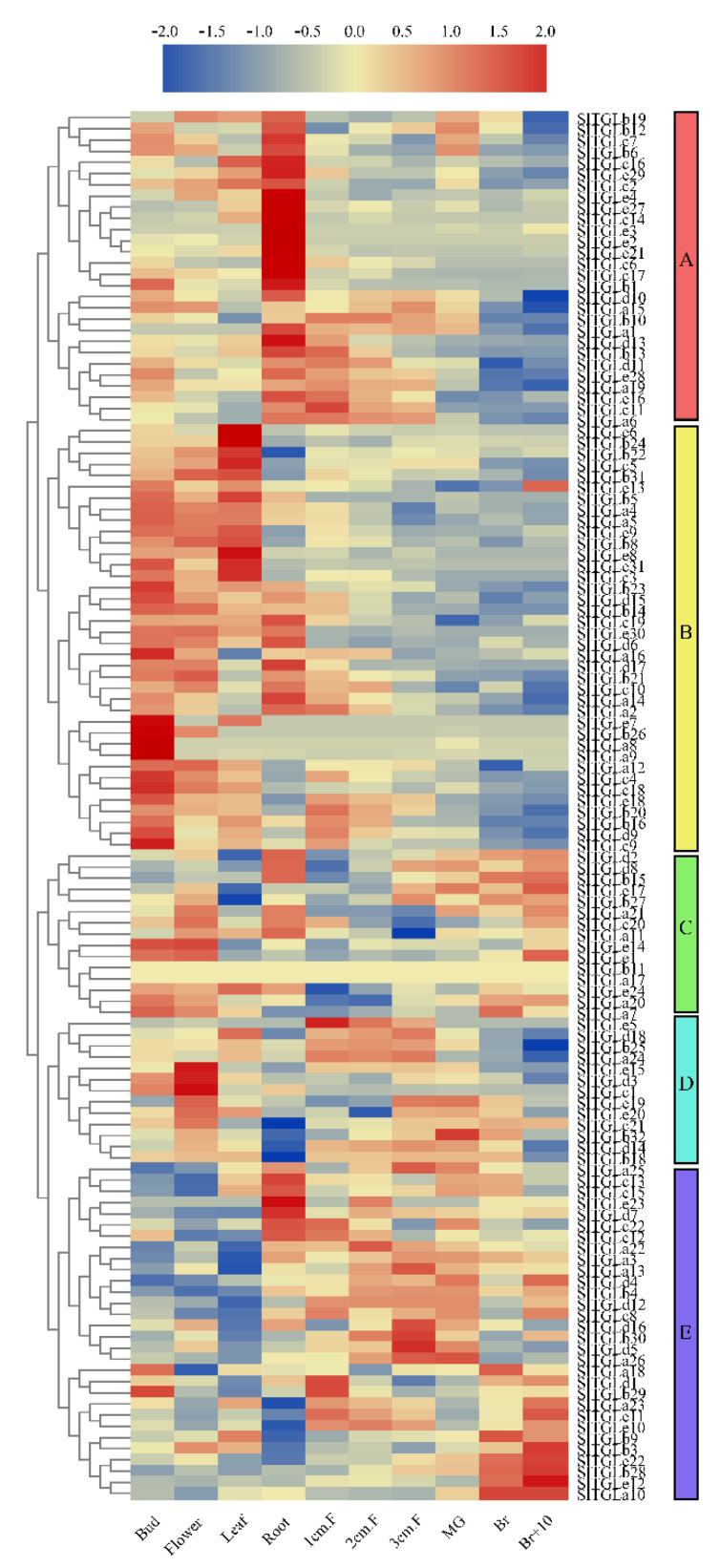
The expression profiles of *SlTGL* genes in cultivated tomato cultivar Heniz 1706 from TFGD. Heatmap of RNA-seq data of Heniz. Fully opened flowers (Flower), Bud, 1-cm fruits (1 cm. F), 2-cm fruits (2 cm. F), 3-cm fruits (3 cm. F), mature green fruits (MG), breaker fruits (Br), fruits at 10 days after breaker stage (Br+10), roots (Root), leaves (Leaf) in the middle. Heatmaps of gene expression profiles were generated using TBtools [45].

**Figure 6 ijms-22-01387-f006:**
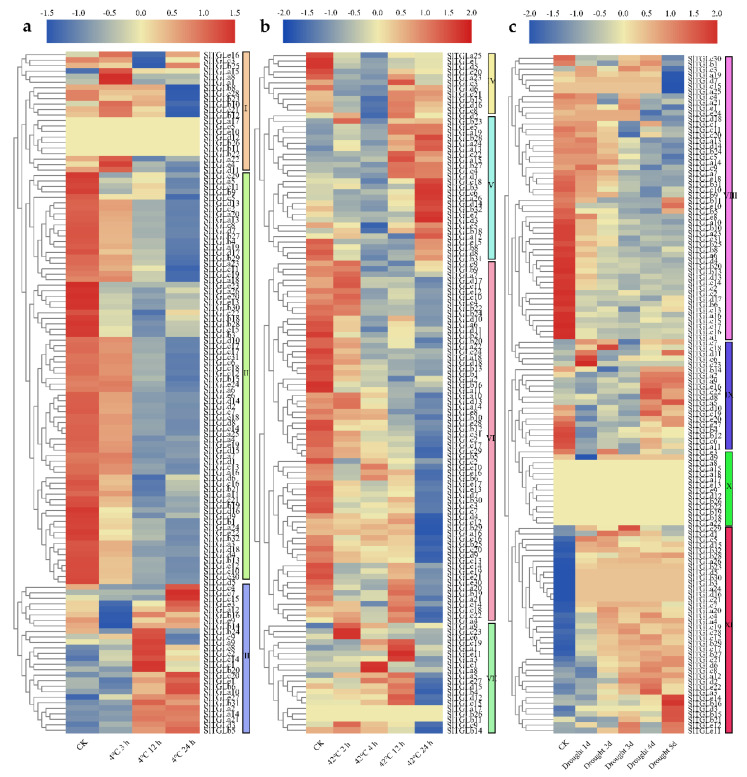
The expression profiles of *SlTGL* genes in chilling (**a**), heat (**b**), and drought (**c**) stress of *S. lycopersicum* cv. M82. (**a**–**c**) Heatmap of RNA-seq data of M82. Due to the differences in expression under different stresses, we re-clustered 129 genes according to their expression levels, and named the re-clustered groups from I to XI. Heatmaps of gene expression profiles were generated using TBtools [45]. Four-week-old tomato plants were treated with different abiotic stresses and leaf samples were collected for the RNA-seq analysis. CK refers to plants before stress treatments in each set. In the color gamut, the beige part represents the same amount of expression.

**Figure 7 ijms-22-01387-f007:**
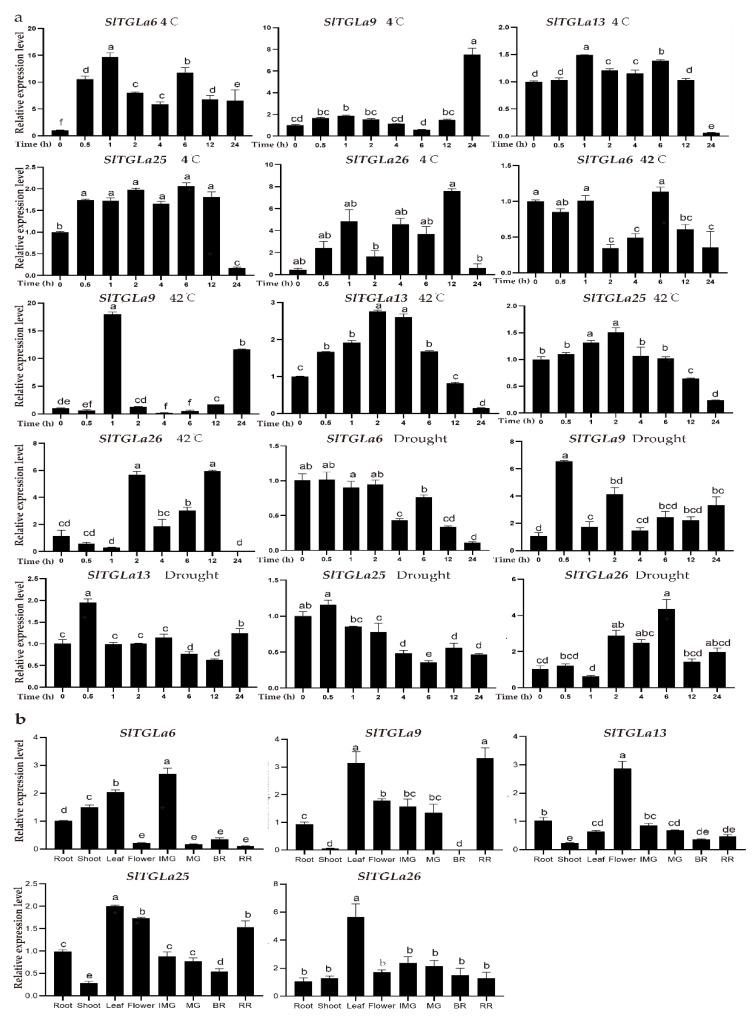
The qRT-PCR analysis of relative expression levels of *SlTGL* genes in 2-week old seedlings under abiotic stresses (**a**) and different organs of 3-month old plants (**b**). IMG, Immature fruit; MG, Mature green fruit; BR, Breaker stage fruit; RR, Red ripe stage fruit. Transcript analysis was performed at 0.5–24 h. The x-axis was time (h). Error bars indicate standard deviation (n = 3). Different letters above bars indicate significant differences in mean values under abiotic stresses or in different organs (*p* < 0.05, one-way ANOVA analysis [Duncan’s multiple range test]).

**Figure 8 ijms-22-01387-f008:**
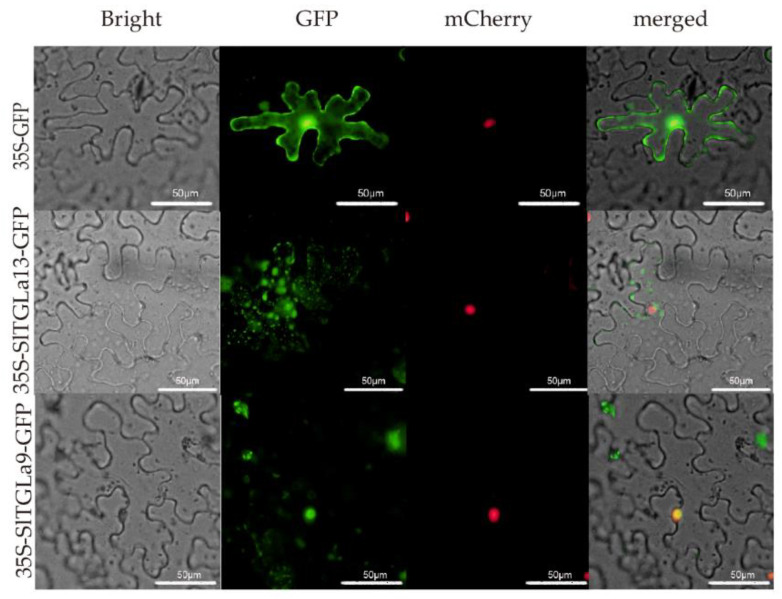
Subcellular localization of two GFP-fused SlTGL proteins. mCherry is a red fluorescent protein (RFP) fused with the nucleus marker, indicating nuclei. Merge indicates merger between the GFP and mCherry fluorescence images. Scale bars, 50 µm.

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
