# Peer review of "Identification, Classification, and Expression Analysis of the Triacylglycerol Lipase (TGL) Gene Family Related to Abiotic Stresses in Tomato"

_ijms, 2021, doi:10.3390/ijms22031387_

Round 1
Reviewer 1 Report
MS titled "Identification, classification, and expression analysis of the triacylglycerol lipase (TGL) gene family related to abiotic stresses in tomato" by Wang et al. gives information regarding the genome-wide identification and characterization of the TGL gene family in tomato (Solanum lycopersicum L.). MS contains mostly in silico analysis of TLG genes and proteins. Authors used available transcroptomic data from the Tomato Functional Genomics Database (TFGD) and performed some own gene expression experiments. Results on own experimental data are confusing and their presentation needs to be improved. Some data are missing (Fig 7b). Authors presented results of NaCl stress and did not even mention that. In Disscusion section there is no word about own experimental results. All of my questions and critisc are available in attached file. I suggest to improve MS before considering for publication.

Reviewer 2 Report
This manuscript presents a study in which available genomic and transcriptomic data in tomato is mined to identify a family of 129 triacylglycerol lipase genes, and to characterize this family relative to gene structure, derivation of family size, and potential responses to developmental and environmental cues by promoter element analyses. In addition, gene expression patterns of the TLG family in plants exposed to different was garnered from a whole transcriptome analysis, however this analysis was not well described in the Results or Methods. Gene expression of five TGL genes within subfamily “a” were further characterized under different stress conditions. This manuscript offers an initial description of this gene family in tomato and provides insights into potential, testable roles of these genes in different developmental processes and in response to different environmental stresses. The study does not functionally characterize these lipases nor does it confirm their roles in protecting against stress. There are several aspects of this study and the presentation of the results that could be enhanced. These key points are presented below, followed by a list of specific comments.
Key Points:
- Segmental duplications: Although the text considers and reports that the size of the TGL family is due to segmental duplications, it is unclear whether the authors studied whether the locations of these TGLs are in known regions of chromosomal segmental duplication that have been previously reported. It would be useful to clarify this topic both in the Results and A more specific comment is provided below.
- Rationale for studying five of the 129 TGLs: Rationale should be provided as to why the five genes (SlTGLa6, SlTGLa9, SlTGLa13, SlTGLa25, and 200 SlTGLa26) were selected for wet-lab experimentation. All five of these genes are from subfamily “a”. While two of these genes were not represented in the TFGD (and so it perhaps make sense to study them), it is not clear why the other were chosen or why subfamily “a” was the focus instead of selecting genes from each of the five subfamilies. Some rationale needs to be provided for this choice and discussed.
- “Our laboratory RNA-seq data”: This manuscript presents an RNAseq analysis of tomato plants exposed to different biotic and abiotic stresses. However, the experiment is not fully presented in this paper, and is instead referred to modestly as “our laboratory RNA-seq data”. It seems that the authors have an excellent opportunity to dig more deeply into these results, and perhaps this is intended for a different manuscript. Regardless, the nature of this transcriptome analysis should be more carefully described and could become a central focus and highlight of the manuscript. Also, it seems that the sequencing data may not yet be deposited into a public database. The methods should reference a pubic deposition of this dataset.
- Discussion: Much of the discussion is a summary of the results (e.g. lines 317-329 and elsewhere) and often does not discuss the results relative to known literature. For example, a comparison of the lipase families in tomato vs. Arabidopsis might be informative, particularly in relation to family size, known functions in response to environmental stress, etc. The authors might also consider a deeper discussion about their conclusion of purifying selection of these lipases and potential implications
Specific comments:
Line 22: Segment duplication analysis is a method, however this sentence seems to say that segmental duplications were the driving force underlying expansion the TGL family in tomato. Remove “analysis” and modify sentence.
Lines 55-58: This definition of triacylglycerols comes late, as TAGs are first presented in the first two paragraphs of the Introduction. Define TAGs in this way at its first introduction earlier in the section instead of defining here.
Line 59: There is a small nod to tomato here that seems out of place here and is distracting. It could likely be removed. The same idea is presented in lines 86-88, where it fits nicely.
Lines 66-67: The sentence “Therefore, TGLs play a vital role in seed germination and seedling development.” seems overstated based on the information presented in this section.
Line 69: In the phrase “Although TAG does not accumulate under normal conditions in plants…”, does “plants” refer to vegetative tissues of plants? This sentence should be clarified, as TAGs do naturally accumulate in some plant tissues (e.g. seeds, embryos, etc.).
Line 91: “Chromosome location” could be interpreted as chromosome mapping, which was not conducted here. “Duplication analysis” is difficult to interpret and perhaps “analysis of gene duplication events” would be more clear.
Section 2.1: This section and others include detailed methods for the bioinformatic analysis (e.g. lines 102-104; lines 168-170) . Much of these methods are also detailed in the Methods. It would be easier to read if the methods text was removed from this section and only the results that were obtained from these methods are reported.
Line 131: The long list of percentages is difficult to parse. Consider instead incorporating this information into Figure S2.
Lines 152-154: For segmental duplication analyses, it is unclear whether these inferences of segmental duplication are relative to known blocks of segmental duplication in tomato, as described in Song et al. (2012) Whole Genome Duplication of Intra- and Inter-chromosomes in the Tomato Genome, Journal of Genetics and Genomics, 39: 361-368. It seems that this would be an important aspect to add to this study. Also, there are examples of tandem duplications in plants that include multiple genes. For at least the duplications on Chromosome 2, is this possible?
Lines 193-194: In this sentence it is stated that in Figure 5 there are seven clustered groups. However, Figure 5 shows five groups as before (a-e). This text needs to be corrected/clarified.
Line 196: In “…in group D were expressed >2 fold in the leaves”, this is in comparison to what?
Lines 219-220: The RNA-seq data need to be described here and not referred to as “our laboratory RNA-seq data”. If this data is already published it should be cited. If not, the manuscript needs to report the ID of the dataset deposited into a public database. Here (or in the legend of Figure 6) it should state the age of the plants and the tissue that was used for RNAseq.
Lines 223-224: In presenting the results associated with Figure 6, there is no mention of Group III genes, which show increased expression after cold treatment.
Line 226: It is stated that “results showed that these five genes had an increasing trend under chilling stress”. Are these trends statistically significant? Results of statistical analyses should be presented in Figure 7.
Lines 244-246: It seems that this list of genes is represented by group X, and perhaps could be replaced by “Group X”.
Line 304: It is unclear why the percentages of genes in subfamilies a and c with transmembrane domains suggests an important role in hydrolyzing TAGs. This conclusion should be more thoroughly discussed.
Line 330: It is concluded that “it was demonstrated that TGL is a multifunction protein” in this study. It is unclear how the data presented demonstrates multifunctionality of the triacylglycerol lipase protein. This protein seems to have a single function as a lipase, and no experiments were reported that test functionality per se. Perhaps the authors are suggesting that different TGLs in this large family can respond to a variety of cues, as suggested by promoter elements and expression analysis under different abiotic stresses. This statement should be carefully and clearly crafted.
Line 337: The transcriptome data generated for this manuscript should be more clearly presented, and brief details included for the “primary analysis”. Also, public database IDs should be provided for these transcriptome datasets.
Lines 395-401: This section in particular needs to be read for clarity, grammar and word choice. It is very difficult to read and interpret. Also, it is unclear what is meant by “amplified from “AC” were”. Methods for co-expression these constructs in tobacco should be included or a reference should be cited.
Figure 1. Is there an outgroup for this phylogenetic tree? Also, consider coloring the names of the two Arabidopsis genes in a different color so they are easily recognized amongst the tomato genes. It would also be helpful to label each colored wedge in the tree with the subfamily letter designation (a-e).
Figure 2. In the figure legend, the meaning of “Details of groups are shown in different colors.” is unclear. Does this refer to the different groups (a-e) being colored as in Figure 1? Bootstrap values should be included in the tree, as was done for Figure 1.
Figure 6. What are the roman numerals I-XI? The text mentions seven groups. CK should be defined in legend. For panel C, the legend might note why all of the gene expression in group 10 is beige.
Figure 7. Gene names and axis labels are very small and difficult read. Results of statistical analysis (e.g. Tukey’s tests) and connecting letters placed above or within the bars could be included. For the salt treatment, were samples not collected at the 6 hour time point? This should be noted in the legend. The legend provides information for a panel B, however no panel B is shown for this figure. The legend should be clarified as to whether the data presented are from 2-week old seedlings or 3-month old plants. If from seedlings, were whole seedlings used for RNA extraction?
Table S1. The numbers reported in the Subcellular Localization column should be defined in a footnote. Also, the heading is misspelled, Subcelular should be Subcellular.
Table S3. There are many empty entries in the “Function of the cis-acting element” and “Type of cis elements” columns. Entries should be added for each cis-acting element. If not yet defined, a NA or Not known or a similar phrase could be added.
Figure S2. The meaning of “Zero or one occurrence of a contributing motif site per sequence” in the legend is unclear and should be reworded for clarity.
Round 2
Reviewer 1 Report
Revised version of MS is partially improved. Authors completed Figure 7. However I do not see any word on salinity stress impact to selected gene transcript. I suggest to delete NaCl results or describe them in section 2.6 as all other abiotic stresses impacts were described.
After this minor revision I recommend MS for publication.

Author Response
Dear Editor and Reviewer,
We sincerely thank you for your assistance and hard work to provide comments/suggestions to improve our manuscript entitled “Identification, classification, and expression analysis of the triacylglycerol lipase (TGL) gene family related to abiotic stresses in tomato”. We have revised our manuscript accordingly and changes throughout the manuscript were highlighted in red color. Below are our responses to the comments/suggestions raised by the reviewer.
Sincerely yours
Jianhua Zhu and Xiangqiang Zhan
Point-by-point responses to reviewers’ comments/suggestions
Reviewer 1
Revised version of MS is partially improved. Authors completed Figure 7. However I do not see any word on salinity stress impact to selected gene transcript. I suggest to delete NaCl results or describe them in section 2.6 as all other abiotic stresses impacts were described.
After this minor revision I recommend MS for publication.
Our response: We thank this reviewer for the valuable comments about our manuscript. We have removed the gene expression data after salinity stress presented in the original Figure 7a following your suggestion. We have also revised the other parts of the manuscript according to your comments and we hope our manuscript is now ready for publication in this journal.
Question 1: You should point out that transcript analysis was performed at 0.5-24h.x-axis is time (h).
Our response: Thank you for your suggestion. According to your suggestion, we have re-written the legend of Figure 7. Please see lines 240-241 in the revised manuscript for details.
Question 2: Figure needs to be self-explanatory, so please explain abbreviations IMG, MG, BR, and RR.
Our response: Thank you for your valuable advice. According to your suggestion, we have modified the legend of Figure 7b and added a description of the abbreviations of IMG, MG, BR, and RR. Please see lines 239-241 in the revised manuscript for details.
Question 3: Again, you did not explain any word on salinity stress. Please delete salinity stress from the figure7a or describe what is going on with certain transcripts under salinity stress.
Our response: Thank you for your careful reading of our manuscript. The qPCR result of salt treatment is our supplementary data to illustrate that SlTGLs might respond to abiotic stress, and removal of these results will not significantly affect the overall strength of our article. Therefore, we have removed the data related to salinity stress in Figure 7 according to your suggestion. Please see Figure7 in the revised manuscript for details.
Reviewer 2 Report
In this revised manuscript, the authors have clarified the Methods, Results and Figure legends and enhanced readability. The inclusion of statistical analysis also improves the manuscript. However, this reader has concerns regarding presentation of the RNA-seq experiment and of the segmental duplication analysis.
Key points:
1. Segmental duplications: I appreciate that the authors reviewed the publication on segmental duplications in tomato. Although they were unable to use this previously published data to gain insights into whether the TGLs were in known segmental duplications, it still seems some clarification and additional analysis is needed. Data is not presented that shows evidence of large-scale segmental duplications with co-linear genes (including one or more TGLs, but also other genes). It seems such evidence would be required to conclude that segmental duplications underlie the size of the TGL family. This would require the reporting of a larger analysis of the sequences flanking the TGLs to show co-linearity in segments. This cannot be inferred from the current Figure 3.
2. RNA-seq experiment: The rationale for this experiment and a brief description of it should be provided in the Results. In the current version, there is no introduction of this experiment in the Results section and the reader must infer the experiment and its purpose from the legend of Figure 6.
Specific comments:
Line 157: The sentence needs to be completed, “…genes in tomato were under purifying.” Perhaps it is meant to say “under purifying selection”.
Line 194: It seems that Figure 7b is cited before Figure 6.
Lines 209-210: There is no introduction of the RNA-seq analysis that is presented in this manuscript. From the sentence, “Based on the primary analysis of our RNA-seq data…” this reader’s reaction was to wonder to what RNA-seq data the authors were referring. This section needs to introduce the RNA-seq experiment. Only a small description and rationale can be found in the legend of Figure 6.
Lines 215-216: I appreciate that the authors responded to my queries regarding why these five genes were selected. However, some sort of rationale also needs to be provided in the text as to why these five genes were selected, all from group “a”.
Line 374: What sequence processing and data analysis methods were used?
Figure 7a. The authors might re-assess the connecting letters reported for the SlTGLa6 panel. It is surprising that the first two bars are not statistically different (they are both labeled with a letter “d”).
Author Response
Dear Editor and Reviewer,
We sincerely thank you for your assistance and hard work to provide comments/suggestions to improve our manuscript entitled “Identification, classification, and expression analysis of the triacylglycerol lipase (TGL) gene family related to abiotic stresses in tomato”. We have revised our manuscript accordingly and changes throughout the manuscript were highlighted in red color. Below are our responses to the comments/suggestions raised by the reviewer.
Sincerely yours
Jianhua Zhu and Xiangqiang Zhan
Point-by-point responses to reviewers’ comments/suggestions
Reviewer 2
In this revised manuscript, the authors have clarified the Methods, Results, and Figure legends, and enhanced readability. The inclusion of statistical analysis also improves the manuscript. However, this reader has concerns regarding the presentation of the RNA-seq experiment and of the segmental duplication analysis.
Our response: We thank this reviewer for the valuable comments about our manuscript. We have revised the manuscript according to your comments and we hope our manuscript is now ready for publication in this journal.
Question 1: Segmental duplications: I appreciate that the authors reviewed the publication on segmental duplications in tomato. Although they were unable to use this previously published data to gain insights into whether the TGLs were in known segmental duplications, it still seems some clarification and additional analysis is needed. Data is not presented that shows evidence of large-scale segmental duplications with co-linear genes (including one or more TGLs, but also other genes). It seems such evidence would be required to conclude that segmental duplications underlie the size of the TGL family. This would require the reporting of a larger analysis of the sequences flanking the TGLs to show co-linearity in segments. This cannot be inferred from the current Figure 3.
Our response: Thank you for your valuable suggestion. We have followed the introduction of the MCScanX software package (DOI: 10.1093/nar/gkr1293, see [41] in the revised manuscript) for data analysis with TGL gene family in tomato. There are many research reports (including example DOIs: 10.1186/s12864-020-06828-z, 10.1186/s12864-020-07069-w and 10.3389/fpls.2020.569838) which conclusions were made with protein-coding gene sequences only. Although we agree that the gene-flanking sequences mentioned in your kindful suggestions should be useful for identification and comparison with collinear blocks (including homologous gene families and their flanking sequences) in genomes of different species, especially in genomes of more than two relative species, however, it seems to be unnecessary in this manuscript, which is focusing on the TGL gene family in tomato only, thus, we would like to keep the original contents.
Question 2: RNA-seq experiment: The rationale for this experiment and a brief description of it should be provided in the Results. In the current version, there is no introduction of this experiment in the Results section and the reader must infer the experiment and its purpose from the legend of Figure 6.
Our response: Thank you for your valuable advice. According to your suggestion, we have added a brief introduction of the RNA-seq data in the Result. Please see lines 209-214 in the revised manuscript for details.
Question 3: Line 157: The sentence needs to be completed, “…genes in tomato were under purifying.” Perhaps it is meant to say “under purifying selection”.
Our response: Thank you very much to point out this information. According to your suggestion, we have re-written the relevant part to “genes in tomato were under purifying selection”. Please see lines 154-156 in the revised manuscript for details.
Question 4: Line 194: It seems that Figure 7b is cited before Figure 6.
Our response: Thank you for your careful reading of our manuscript. qRT-PCR results in Figure 7b are validation of transcriptome of SlTGLs in different organs presented in Figure 5. So it is reasonable to cite Figure 7b before Figure 6 (transcriptome of SlTGLs under abiotic stress conditions).
Question 5: Lines 209-210: There is no introduction of the RNA-seq analysis that is presented in this manuscript. From the sentence, “Based on the primary analysis of our RNA-seq data…” this reader’s reaction was to wonder to what RNA-seq data the authors were referring. This section needs to introduce the RNA-seq experiment. Only a small description and rationale can be found in the legend of Figure 6.
Our response: Thank you for your suggestion. We have added a brief introduction of the RNA-seq analysis to provide the rationale for this experiment and please lines 209-214 in the revised manuscript for details.
Question 6: Lines 215-216: I appreciate that the authors responded to my queries regarding why these five genes were selected. However, some sort of rationale also needs to be provided in the text as to why these five genes were selected, all from group “a”.
Our response: Thank you for your careful reading of our manuscript. We explained the rationale why we choose these five genes for further analysis by qRT-PCR. Please see lines 191-195in the revised manuscript for details.
Question 7: Line 374: What sequence processing and data analysis methods were used?
Our response: Thank you for your suggestion. The transcriptome data used in this manuscript is obtained through the next generation sequencing technology with Illumina HiSeq2500. RNA-Seq data were analyzed by the TopHat and Bowtie2 programs. We used the Cuffdiff, a program within Cufflinks (http://cufflinks.cb-cb.umd.edu/), for differential gene expression analysis. Please see lines 378-381 in the revised manuscript for details.
Question 8: Figure 7a. The authors might re-assess the connecting letters reported for the SlTGLa6 panel. It is surprising that the first two bars are not statistically different (they are both labeled with a letter “d”).
Our response: Thank you for your careful reading of our manuscript. We are sorry again for this mistake. There is indeed a difference between the first two bars, which we have corrected in the revised Figure 7a.